# Molecular Imaging and Preclinical Studies of Radiolabeled Long-Term RGD Peptides in U-87 MG Tumor-Bearing Mice

**DOI:** 10.3390/ijms22115459

**Published:** 2021-05-21

**Authors:** Wei-Lin Lo, Shih-Wei Lo, Su-Jung Chen, Ming-Wei Chen, Yuan-Ruei Huang, Liang-Cheng Chen, Chih-Hsien Chang, Ming-Hsin Li

**Affiliations:** 1Division of Isotope Application, Institute of Nuclear Energy Research, Taoyuan 32546, Taiwan; loweilin@iner.gov.tw (W.-L.L.); hwulab@iner.gov.tw (S.-W.L.); totatakimo@iner.gov.tw (S.-J.C.); brian0512@iner.gov.tw (M.-W.C.); yuanruei@iner.gov.tw (Y.-R.H.); lcchen@iner.gov.tw (L.-C.C.); 2Department of Biomedical Imaging and Radiological Sciences, National Yang Ming Chiao Tung University, Taipei 11221, Taiwan

**Keywords:** ^111^In-DOTA-EB-cRGDfK, indium-111, nanoSPECT/CT, pharmacokinetics

## Abstract

The Arg–Gly–Asp (RGD) peptide shows a high affinity for α_v_β_3_ integrin, which is overexpressed in new tumor blood vessels and many types of tumor cells. The radiolabeled RGD peptide has been studied for cancer imaging and radionuclide therapy. We have developed a long-term tumor-targeting peptide DOTA-EB-cRGDfK, which combines a DOTA chelator, a truncated Evans blue dye (EB), a modified linker, and cRGDfK peptide. The aim of this study was to evaluate the potential of indium-111(^111^In) radiolabeled DOTA-EB-cRGDfK in α_v_β_3_ integrin-expressing tumors. The human glioblastoma cell line U-87 MG was used to determine the in vitro binding affinity of the radiolabeled peptide. The in vivo distribution of radiolabeled peptides in U-87 MG xenografts was investigated by biodistribution, nanoSPECT/CT, pharmacokinetic and excretion studies. The in vitro competition assay showed that ^111^In-DOTA-EB-cRGDfK had a significant binding affinity to U-87 MG cancer cells (IC_50_ = 71.7 nM). NanoSPECT/CT imaging showed ^111^In-DOTA-EB-cRGDfK has higher tumor uptake than control peptides (^111^In-DOTA-cRGDfK and ^111^In-DOTA-EB), and there is still a clear signal until 72 h after injection. The biodistribution results showed significant tumor accumulation (27.1 ± 2.7% ID/g) and the tumor to non-tumor ratio was 22.85 at 24 h after injection. In addition, the pharmacokinetics results indicated that the ^111^In-DOTA-EB-cRGDfK peptide has a long-term half-life (T_1/2__λ__z_ = 77.3 h) and that the calculated absorbed dose was safe for humans. We demonstrated that radiolabeled DOTA-EB-cRGDfK may be a promising agent for glioblastoma tumor imaging and has the potential as a theranostic radiopharmaceutical.

## 1. Introduction

Cancer is the second leading cause of death in the world. According to statistics from the World Health Organization (WHO), about 9.6 million people die from cancer each year, and on average, 1 in 6 people die from cancer. In recent years, integrin-mediated biological activity targeted against cancer has been improving. Mammals have 24 different integrins, consisting of 18 α-subunits and 8 β-subunits, which are cell adhesion receptors connected to the extracellular matrix (ECM) and other cells [1]. Integrins participate in signal transduction pathways to regulate cell growth, migration, motility, proliferation and survival [2,3]. The α_v_β_3_ integrin is overexpressed in new tumor blood vessels and various tumor cells including neuroblastoma, osteosarcoma, melanoma, glioblastoma, breast cancer and lung cancer [4,5]. Furthermore, integrin plays important roles in cancer proliferation, invasion, survival, metastasis and angiogenesis [6,7,8]. According to the above characteristics, integrin is developed as a suitable target for cancer diagnosis and therapy [9,10,11].

The Arg–Gly–Asp (RGD) peptide is expressed on ECM and recognized by integrin, and shows a high affinity for α_v_β_3_ integrin [12,13]. Since the RGD peptide was first discovered in 1984 [14], plenty of studies have been investigated on peptides containing RGD sequences to accurately treat various tumors [15,16,17]. The RGD-containing peptides can be divided into linear and cyclic forms in structure, and have different modifications for different purposes, such as to increase binding affinity, extend half-life, combine with another drug, etc. Cyclization of RGD can enhance biological activity and significantly improve its selectivity and binding ability to receptors, and the cyclic conformational structure can also prevent proteolysis [18,19]. The cyclic Arg–Gly–Asp–D–Phe–Lys (cRGDfK) sequence can inhibit the adhesion of fibronectin to cells, and the binding affinity to integrin α_v_β_3_ is much higher than the linear RGD peptide sequence [20].

In vivo and non-invasive molecular imaging are being used in the localization of tumors and the staging of cancer. In all molecular imaging modalities, positron emission tomography (PET) and single photon emission computed tomography (SPECT) occupy a specific location, using high-affinity and high-specific molecular radiotracers as imaging probes to visualize and measure physiological processes [21]. Targeting peptides are combined with different radionuclides, diagnostic or therapeutic isotopes, and are often used as radiotracers. The radiolabeled RGD peptide has been studied for cancer imaging and radionuclide therapy [22]. The first radiotracer developed using the RGD sequence concept, such as ^18^F-Galacto-RGD, was used to diagnose head and neck squamous cell carcinoma [23]. Many RGD-derived drugs have entered clinical trials, such as ^68^Ga-NOTA-BBN-RGD (Phase 1), ^18^F-RGD-K5 (Phase 2), ^18^F-Al-NOTA-PRGD2 (Phase 1), and ^18^F-FPPRGD2 (Phase 2), and are used to diagnose different types of cancer. However, there are many potential drug treatments fail to reach effective concentrations due to unfavorable pharmacokinetics. Recently, Chen et al. published a novel EB-RGD for imaging and radiotherapy, consisting of a truncated Evans blue dye (EB) molecule with the binding capacity of albumin, a metal chelate that allows radiolabeling, and RGD peptide that binds to integrin [24]. The micromolar affinity and reversible binding of EB derivatives to albumin extend the half-life of the drug in the blood. The EB-conjugated drug can improve pharmacokinetic properties and prolongs blood circulation. Moreover, EB-RGD may become an effective treatment option for targeted radionuclide therapy (TRT) by labeling with the β-emitting radionuclide lutetium-177 (^177^Lu) [25,26].

In this context, we developed a novel long-circulation integrin-targeted molecule DOTA-EB-cRGDfK based on EB-RGD. To increase tumor accumulation and retention for radioligand therapy, and reduce dosage of radionuclide, we designed and conjugated an EB molecule and DOTA chelator onto RGD peptide and labeled it with indium-111(^111^In). DOTA is a good chelator for labeling different radionuclides, such as ^68^Ga for PET imaging, ^111^In for SPECT imaging, ^177^Lu and ^90^Y for treatment. Conjugated with EB can improve the short half-life of peptides in the body and increase tumor accumulation [24]. To optimize biodistribution and reduce kidney absorption, we explored a new linker that has a longer carbon chain and is more lipophilic than EB-RGD. ^111^In was used because it emits gamma rays (171 and 245 keV), with a long half-life (2.8 days), and is suitable for long-term imaging. Additionally, we investigated the binding specificity, SPECT/CT imaging, pharmacokinetics, and biodistribution of radiolabeled DOTA-EB-cRGDfK. These studies selected U-87 MG human glioblastoma with a high expression of α_v_β_3_ integrin as the model. In addition, we developed two control peptides for comparison with the targeting peptide. DOTA-cRGDfK is the current standard of RGD peptides, which is used to compare the tumor-targeting effects. The purpose of DOTA-EB is to understand whether EB has a binding or modulating effect on tumors. This work demonstrated that radiolabeled DOTA-EB-cRGDfK is a promising agent for glioblastoma imaging and has the potential as a theranostic radiopharmaceutical.

## 2. Results

### 2.1. Radiolabeling with Indium-111

The structure of targeting peptide DOTA-EB-cRGDfK (A) and the two control peptides DOTA-cRGDfK (B) and DOTA-EB (C) are shown in Figure 1. After radiolabeling with ^111^In, the labeling efficiency of these peptides was higher than 95% determined by radio-TLC analysis (Figure 2A). The R_f_ of ^111^In-DOTA-EB-cRGDfK, ^111^In-DOTA-cRGDfK, and ^111^In-DOTA-EB were close to 0, and the R_f_ of free ^111^In was approximately 1. In addition, HPLC analysis showed the radiochemical purity of the ^111^In-DOTA-EB-cRGDfK, ^111^In-DOTA-cRGDfK, and ^111^In-DOTA-EB was greater than 95% (Figure 2B). The retention time of free ^111^In, ^111^In-DOTA-EB-cRGDfK, ^111^In-DOTA-cRGDfK, and ^111^In-DOTA-EB was 4.8, 7.1, 5.9 and 8.2 min, respectively. Because the labeling efficiency of ^111^In-DOTA-EB-cRGDfK was over 95% (96.3 ± 1.4%, *n* = 3), no further purification was performed for the subsequent studies.

### 2.2. In Vitro Competitive Binding Assay

Competitive binding assay with U-87 MG cells revealed specific cell uptake of ^111^In-DOTA-EB-cRGDfK. As the concentration of unlabeled DOTA-EB-cRGDfK or DOTA-cRGDfK increased, the binding ratio of ^111^In-DOTA-EB-cRGDfK decreased, indicating the presence of specific binding affinity (Figure 2C). The half maximal inhibitory concentration (IC_50_) values of DOTA-EB-cRGDfK or DOTA-cRGDfK were obtained from curve fitting, which were 71.7 nM and 35.2 nM, respectively.

### 2.3. In Vitro Stability Study

The radiochemical purity of ^111^In-DOTA-EB-cRGDfK after incubation in rat plasma was 96.8 ± 1.8%, 95.7 ± 2.1%, 96.3 ± 1.9%, 95.8 ± 1.3%, 96.2 ± 1.3%, 96.3 ± 2.0%, 94.2 ± 2.2% at 0 h, 2 h, 4 h, 24 h, 48 h, 72 h and 96 h, respectively. In addition, similar results were shown in normal saline incubation (Table 1). The stability of ^111^In-DOTA-EB-cRGDfK in normal saline or rat serum was greater than 95% within 2 h incubation, and was still greater than 94% after 96 h.

### 2.4. NanoSPECT/CT Imaging

Figure 3 presented nanoSPECT/CT images of U-87 MG xenografts in mice administered with different radiolabeled peptides. After injection of ^111^In-DOTA-EB-cRGDfK, tumor accumulation can be found within 0.5 h, and gradually increased with time and there was still a clear signal until 72 h. ^111^In-DOTA-EB-cRGDfK showed significantly higher tumor uptake than control peptides at all time points by image quantification results (12.36 ± 0.88, 16.59 ± 2.16, 18.66 ± 2.15, 25.09 ± 4.76, 25.49 ± 3.88 and 23.61 ± 2.98% ID/g in tumor at 0.5, 2, 4, 24, 48 and 72 h after injection, respectively). When blocking with 120-fold molar excess of DOTA-EB-cRGDfK, the image showed significantly inhibitive binding of ^111^In-DOTA-EB-cRGDfK in tumors (5.41 ± 1.24, 5.69 ± 1.31, 5.84 ± 0.88, 7.13 ± 2.62, 8.19 ± 2.22 and 7.52 ± 2.35% ID/g in tumors at 0.5, 2, 4, 24, 48 and 72 h after injection, respectively). Control peptide ^111^In-DOTA-cRGDfK showed low tumor uptake and rapid excretion from the body. Comparing with ^111^In-DOTA-EB-cRGDfK, the highest tumor uptake of ^111^In-DOTA-cRGDfK was 2.0 ± 0.5% ID/g at 0.5 h. In addition, ^111^In-DOTA-EB showed relatively obvious radiation signals in non-tumor organs such as muscle and liver. The ratios of tumor to muscle and tumor to liver were 1.8 ± 0.7 and 0.5 ± 0.1 at 0.5 h, respectively. The highest tumor uptake of ^111^In-DOTA-EB was 10.3 ± 4.4% ID/g at 24 h, and it slightly dropped to 9.97 ± 4.37% ID/g at 72 h.

### 2.5. Biodistribution Studies

Biodistribution of ^111^In-DOTA-EB-cRGDfK at 2, 24, 48, 72, and 96 h after intravenous injection is listed in Table 2. The results showed that the top three organs for drug accumulation were the tumor, kidney, and spleen. The radioactivity level at the tumor site reached a peak at 24 h (27.1 ± 2.7% ID/g) and was steadily maintained until 96 h after administration (18.6 ± 2.0% ID/g). The Tumor/Muscle (T/M) ratio of ^111^In-DOTA-EB-cRGDfK was 8.9 at 2 h and reached to 22.85 at 24 h. The highest uptake in the kidney and spleen was 14.4 ± 0.4% ID/g at 24 h and 12.9 ± 2.8% ID/g at 48 h, respectively. Albumin binding ability of EB may cause accumulation in the spleen, and the intense radioactivity seen in the kidneys and bladder suggests ^111^In-DOTA-EB-cRGDfK was cleared by renal excretion. In addition, low levels of radioactivity were observed in the musculoskeletal systems and other organs.

### 2.6. Pharmacokinetic and Excretion Studies

The pharmacokinetic parameters were estimated with the WinNonlin program and summarized in Table 3. The maximum concentration (C_max_) of ^111^In-DOTA-EB-cRGDfK and ^111^In-DOTA-cRGDfK in blood was 13.1% ID/g and 3.74% ID/g. The elimination half-life (T_1/2__λ__z_) of ^111^In-DOTA-EB-cRGDfK and ^111^In-DOTA-cRGDfK was 77.3 h and 17.2 h. The clearance rate (Cl) of ^111^In-DOTA-EB-cRGDfK and ^111^In-DOTA-cRGDfK was 0.426 mL/h and 12.9 mL/h. The area under the time curve (AUC_0→__∞_) of ^111^In-DOTA-EB-cRGDfK and ^111^In-DOTA-cRGDfK was 242% ID/mL×h and 4.02% ID/mL × h. The MRT(_0→__∞_) of ^111^In-DOTA-EB-cRGDfK and ^111^In-DOTA-cRGDfK in blood were 77.2 h and 10.3 h, respectively. These results showed that RGD peptide combined with EB can extend half-life, AUC, and MRT in the body.

The excretion data of ^111^In-DOTA-EB-cRGDfK from U-87 MG tumor-bearing mice are presented in Figure 4. The cumulative radioactivity excreted via urine and feces were 38.7 ± 7.0% ID and 21.6 ± 5.6% ID up to Day 8 post-injection, respectively. After 8 days of the administration, ^111^In-DOTA-EB-cRGDfK was excreted from the body by 60%, and the radioactivity was primarily cleared through the urine, accounting for 64% of the total excretion.

### 2.7. Dosimetry

According to the determination of the residence time in mice, the radiation absorbed dose prediction of ^111^In-DOTA-EB-cRGDfK administered to humans is shown in Table 4. Urinary excretion was assumed to be 37%, biologic half-time was 17.6 h, and intestinal excretion was assumed to be 25%. The radiation absorbed dose was low, of which the first three absorbed doses were lower large intestine (0.622 mSv/MBq), upper large intestine (0.379 mSv/MBq), and kidneys (0.214 mSv/MBq). In other treatment-limiting tissues, such as heart, liver, lungs, ovaries, red marrow and spleen, the absorption values were 0.102 mSv/MBq, 0.187 mSv/MBq, 0.096 mSv/MBq, 0.202 mSv/MBq, 0.08 mSv/MBq, 0.186 mSv/MBq, respectively. The effective dose was 0.201 mSv/MBq.

## 3. Discussion

Radiolabeled peptides that bind to cancer cells with high affinity and specificity may have great potential for both diagnostic imaging and targeted radionuclide therapy. Such peptide molecules are cleared from circulation very quickly. Therefore, radiotherapy requiring high doses and frequent clinical administration results in higher clinical costs and systemic toxicity [27,28]. In this study, we present the preclinical evaluation of a novel ^111^In-DOTA-EB-cRGDfK peptide. It consists of a tumor-targeting peptide cRGDfK, an albumin-binding agent (EB), a modified linker, and a DOTA chelator for radionuclide labeling. The cRGDfK peptide shows specificity towards α_v_β_3_ integrin receptors that are most commonly over-expressed in tumor cells, which can guide peptides to tumor sites [29]. Binding to plasma protein is an effective strategy to increase the pharmacokinetic properties of short-lived molecules [30]. The EB has an affinity to albumin and can improve the pharmacokinetic properties of the peptide.

Because ^111^In is a long-lived radioisotope and can be used to prepare tracers for diagnostic imaging, the radiolabeling of peptides with ^111^In has been used for tumor detection [31,32]. We radiolabeled DOTA-EB-cRGDfK with gamma-emitting radionuclide ^111^In and obtained high radiochemical purity (>95%) without further purification. Additionally, it showed high stability, as the radiochemical purity of ^111^In-DOTA-EB-cRGDfK remained above 94% for at least 96 h. The competitive binding assay showed either DOTA-EB-cRGDfK or DOTA-cRGDfK had specificity to U-87 MG human glioblastoma cell. The IC_50_ of DOTA-cRGDfK (35.2 nM) was almost identical to the previous report [24,33]. Although the results showed that the conjugate of RGD peptide to EB molecule slightly reduced the affinity (IC_50_ = 71.7 nM), it was still at the nM level with good binding specificity. The binding properties of EB to albumin can be used to retain drugs in the blood and be supported by the in vivo results.

NanoSPECT/CT images of ^111^In-DOTA-EB-cRGDfK showed significant tumor signal within 0.5 h, and gradually increased with time, and maintained to 72 h overtime. On the contrary, the signal of ^111^In-DOTA-cRGDfK was not obvious under the same scale bar and was quickly eliminated from the body. Another control group, ^111^In-DOTA-EB, also had significant tumor accumulation and long half-life, but obvious radiation signals could be observed in non-tumor organs within 4 h after administration. According to image quantification results, the highest tumor uptake of ^111^In-DOTA-EB-cRGDfK, ^111^In-DOTA-cRGDfK, and ^111^In-DOTA-EB was 25.5 ± 3.9% ID/g at 48 h, 2.0 ± 0.5% ID/g at 0.5 h, and 10.3 ± 4.4% ID/g at 24 h, respectively. The tumor uptake of ^111^In-DOTA-cRGDfK was consistent with the previous study [24,33]. The accumulation of ^111^In-DOTA-EB in tumors is mainly due to the enhanced permeability and retention (EPR) effect. EB may bind strongly to albumin, extravasates, and remains for a prolonged time in the extravascular space due to the EPR effect of tumors [34]. The tumor accumulation of ^111^In-DOTA-EB-cRGDfK is greater than the sum of ^111^In-DOTA-cRGDfK and ^111^In-DOTA-EB. When the RGD peptide prolongs its retention time in the body, it may increase the chance of tumor targeting, leading to increased tumor uptake; then, competing with excess DOTA-EB-cRGDfK, the inhibitory effect gradually increased with time and reached 71% in 24 h. This indicated ^111^In-DOTA-cRGDfK has specific binding toward U-87 MG solid tumor.

Biodistribution results were consistent with the drug properties shown in the bioimaging results. The maximum accumulation of the drug at the tumor site can reach 27.12 ± 2.70% ID/g, and it remains at 18.55 ± 2.01% ID/g after 96 h administration; then, the maximum tumor to muscle ratio and tumor to blood ratio was 22.85 and 23.61, respectively. These indicate that DOTA-EB-cRGDfK has a good specific binding ability to tumors. The results show that the high non-specific intake in the spleen is high due to the albumin binding affinity of the EB portion. In addition, the radioactivity was seen in the kidneys and urinary bladder suggested renal excretion, which is consistent with the results of the excretion test. The excretion data of ^111^In-DOTA-EB-cRGDfK showed that urinary excretion accounted for 64% of the total excretion, and 60% injection dose was excreted from the body 8 days after administration.

Plasma protein binding can be an effective strategy for improving the pharmacokinetic properties of short-lived molecules [30]. For the pharmacokinetics, the experimental results demonstrated a pharmacologic advantage of ^111^In-DOTA-EB-cRGDfK. The C_max_ of ^111^In-DOTA-EB-cRGDfK in the body is 3.5 times that of ^111^In-DOTA-cRGDfK, and the half-life is 4.5 times longer. Moreover, ^111^In-DOTA-EB-cRGDfK showed slow blood clearance, slow mean residence time, and prolonged retention. The biodistribution and pharmacokinetic results demonstrated the longer retention time in the blood, long-term tumor localization, and high concentration of DOTA-EB-cRGDfK.

In terms of drug toxicity, the first-in-human study of ^64^Cu-NOTA-EB-RGD demonstrated its safety, favorable pharmacokinetic, and dosimetry profile [35]. The DOTA-EB-cRGDfK also showed safe dosimetric calculations through the OLINDA/EXM code, providing estimates of the absorbed doses from animal species to humans. In order to evaluate the potential toxicity of ^111^In-DOTA-EB-cRGDfK in clinical use, we took the recommended intravenous dose (222 MBq) for SPECT imaging of an FDA-approved ^111^In-labeled peptide radiotracer, ^111^In-pentetreotide, as reference. The radiation doses of ^11^In-DOTA-EB-cRGDfK from such an administration for the lower large intestine, kidney, and liver were 0.138 Gy, 0.048 Gy, and 0.042 Gy, respectively. Compare to the tolerance of normal tissues to radiation doses are 25 Gy, 20 Gy, and 8 Gy, respectively [36], we found that the radiation absorbed dose was low and does not affect the human body. This shows the safety of the drug and is also conducive to future related trials of labeling for therapeutic radionuclides.

The novel DOTA-EB-cRGDfK peptide presented several advantages in U-87 MG human glioblastoma. First, it shows a high tumor uptake in vivo biodistribution (27.12 ± 2.70% ID/g), and it is higher than dimeric, trimeric, and tetramer c(RGD) radioligands [37,38,39], and the tumor imaging quantification of ^111^In-DOTA-EB-cRGDfK is 50% higher than ^64^Cu-NMEB-RGD (25.09 ± 4.76% ID/g and 16.64 ± 1.99% ID/g at 24 h after injection) [24]. In addition, the tumor uptake of ^18^F-BPA is 1.44 ± 0.44% ID/g, which is usually used in boron neutron capture therapy (BNCT) [40]. The tumor absorption of ^111^In-DOTA-EB-cRGDfK is more than 20 times that of ^18^F-BPA. Second, ^111^In-DOTA-EB-cRGDfK shows favorable pharmacokinetics. The radiolabeled peptides usually showed fast blood clearance, but the T_1/2__λ__z_ of ^111^In-DOTA-EB-cRGDfK was 77.3 h. The extended half-life may increase the fully useful dosage. Third, DOTA is a good chelator for labeling different radionuclides, such as ^111^In for SPECT imaging and ^177^Lu for treatment. DOTA-EB-cRGDfK shows the possibility of being a theranostic radiopharmaceuitcal. On the other hand, blood-brain barrier (BBB) may be considered in the future application. Radiolabeled peptides may pass the barrier because BBB breakdown is common in high-grade gliomas and brain metastases [41,42]. Hematotoxicity may be concerned due to albumin binding ability, and further investigations with kidneys and intestine toxicities of administration are warranted. In addition, we have other experiments with DOTA-EB-cRGDfK in different cell types in progress. Experiments therapeutic radionucleotide ^177^Lu labeling are also ongoing.

## 4. Materials and Methods

### 4.1. Cell Culture and Tumor Model

The human glioblastoma cell line, U-87 MG (stock number: BCRC 60360), was obtained from Bioresource Collection and Research Center. Cells were grown in Minimum Essential Medium (MEM; Gibco, Grand Island, NY, USA) supplemented with 10% (*v*/*v*) fetal bovine serum (FBS; Gibco, Gaithersburg, MD, USA) and 1% antibiotic antimycotic solution (Sigma-Aldrich, St. Louis, MO, USA) at 37 °C under 5% CO_2_ incubation. Female non-obese diabetic, severe combined immunodeficiency (NOD/SCID) mice were obtained from BioLASCO Taiwan Co., Ltd. The 6-week-old mice were housed in a 12 h light cycle at 22 °C, with food and water provided ad libitum. 2 × 10^6^ U-87 MG cells resuspended in 100 μL of phosphate-buffered saline (PBS; Gibco, Gaithersburg, MD, USA) were injected subcutaneously into each nude mouse. The tumor model was studied when the tumor volume reached 300–500 mm^3^. Animal protocols were approved by the Institutional Animal Care and Use Committee at the Institute of Nuclear Energy Research, Taoyuan, Taiwan (approval ID: 107169; approval date:13 February 2018).

### 4.2. Radiolabeling with Indium-111

The targeting peptide DOTA-EB-cRGDfK, DOTA-cRGDfK, and DOTA-EB were purchased from Ontores Biotechnologies (Zhejiang, China). Indium [^111^InCl_3_] chloride (gamma emitter) was generated from INER (Lung-Tan, Taiwan). For indium-111 labeling, 30 μg targeting peptide and 222 MBq ^111^InCl_3_ were mixed in 1 M sodium acetate (pH 6.0) to a final volume of 300 μL. After incubation for 15 min at 95 °C and 500 rpm shaking in a thermal controller, the radiolabeling product was obtained without further purification. Product quality was analyzed by instant thin layer chromatography (iTLC) and high-performance liquid chromatography (HPLC).

The iTLC method was used on the glass microfiber chromatography paper impregnated with a silica gel (Agilent Technologies, Santa Clara, CA, USA), whereas the mobile phase was used as 0.1 M citric acid/0.1 M sodium citrate (*v*/*v* = 2.1/7.9) buffer. Then, the sheets were measured using a radioactive scanner (AR-2000 radio-TLC Imaging Scanner, Bioscan, France), and the relative front (Rf) value of radiochemical was calculated. Rf value is defined as the ratio of distance traveled by the component to the distance traveled by the solvent front from the sample spot. HPLC-analysis was performed using UV detection at 220 nm and a radio detector with a Zorbax SB-C18 column (Agilent Technologies, Santa Clara, CA, USA). The flow rate was 0.8 mL/min with the gradient mobile phase going from 80% A buffer (0.1% TFA in water) and 20% B buffer (0.1% TFA in acetonitrile) to 60% B buffer within 10 min.

### 4.3. In Vitro Competitive Binding Assays

U-87 MG cells were seeded into 24-well plates at a density of 1 × 10^5^ cells per well and incubated overnight. Different concentrations ranging from 0.01 nM to 5000 nM of DOTA-EB-cRGDfK or DOTA-cRGDfK were added to the wells. U-87 MG cells were then incubated with 10 nM ^111^In-DOTA-EB-cRGDfK for 4 h at 37 °C. Cells were washed once with 0.5 mL PBS and were removed from each well by 0.25% trypsin-EDTA (Gibco, Gaithersburg, MD, USA). The cell suspensions were collected and measured by using a PerkinElmer 2480 Automatic Gamma Counter (PerkinElmer, Waltham, MA, USA). Binding results were expressed as a percent of total counts, and IC_50_ values were calculated using SigmaPlot 12.5 (Systat Software, Inc., San Jose, CA, USA).

### 4.4. In Vitro Stability Study

In vitro stability of ^111^In-DOTA-EB-cRGDfK was evaluated by incubation with normal saline (in 1:1 vol ratio) or rat plasma (in 1:19 vol ratio) at room temperature. The radiochemical purity was determined by iTLC analysis as previously described [43]. At desired times (0, 2, 4, 24, 48, 72, and 96 h), 1 μL was using for iTLC on the glass microfiber chromatography paper impregnated with a silica gel (Agilent Technologies, Santa Clara, CA, USA), whereas the mobile phase was used as 0.1 M citric acid/0.1 M sodium citrate (*v*/*v* = 2.1/7.9) buffer. Then, the sheets were measured using a radioactive scanner (AR-2000 radio-TLC Imaging Scanner, Bioscan, France).

### 4.5. NanoSPECT/CT Imagings

The procedure for nanoSPECT/CT imaging has been previously described [43]. Each mouse was tail-vein injected with about 18.5 MBq and 2.5 μg (50–100 μL) of radiolabeled ^111^In-DOTA-EB-cRGDfK, ^111^In-DOTA-cRGDfK, or ^111^In-DOTA-EB. For the blocking study, animals were pre-treated with a 120-fold molar excess (300 μg) DOTA-EB-cRGDfK by tail vein injection 10 min ago. SPECT images and X-ray CT images were acquired using a nanoSPECT/CT^®^ plus scanner system (Mediso Medical Imaging Systems; Budapest, Hungary). The mice were anesthetized with 1–2% isoflurane during the imaging acquisition. The imaging acquisition was accomplished at 60 s per frame. The energy window was set at 171 and 245 KeV ± 10%, the image size was set at 256 × 256, and the field of view of 60 mm × 100 mm. For image reconstruction, the HiSPECT and Nucline software were used for the SPECT and CT images, respectively. The InVivoScope software was used for the fusion of SPECT and CT images. The image quantification of the region of interest (ROI) was acquired by PMOD v. 3.3 (Zürich, Switzerland). The SPECT images were presented on a scale of 2.5% ID/g to 25% ID/g.

### 4.6. Biodistribution, Pharmacokinetic and Excretion Studies

Thirty U-87 MG tumor-bearing mice (five mice per group) received an intravenous injection of about 1.85 MBq and 0.25 μg of ^111^In-DOTA-EB-cRGDfK for biodistribution study. They were sacrificed by CO_2_ asphyxiation at 2, 4, 24, 48, 72, and 96 h. At each time point, the organs of interest were sampled and whole organs collected where possible. The samples were rinsed in saline, blotted dry, weighed, and then counted using a PerkinElmer 2480 gamma counter (PerkinElmer, Waltham, MA, USA). Data were expressed as the percentage injected dose per gram of tissue (% ID/g).

For the pharmacokinetic study, five U-87 MG tumor-bearing mice received an intravenous injection of about 1.85 MBq and 0.25 μg of ^111^In-DOTA-EB-cRGDfK or ^111^In-DOTA-cRGDfK. Blood samples (10 μL) were collected by heart puncture under 2% isoflurane anesthesia at 0.083, 0.5, 2, 4, 24, 48, 72, 96, and 168 h. The radioactivity of blood samples was measured by PerkinElmer 2480 gamma counter and expressed as the percentage injected dose per gram (% ID/g). Data was used to determine the pharmacokinetic parameters by noncompartmental analysis (NCA) and the analysis software was WinNonlin 7.0 (Pharsight Corporation, Mountain View, CA, USA). Parameters, including terminal half-life (T_1/2__λ__z_), maximum concentration (Cmax), total body clearance (Cl), area under curve (AUC), and mean residence time (MRT) were determined. Pharmacokinetic parameters associated with the terminal phase were calculated using the best-fit method to estimate the terminal half-life.

For excretion studies, five U-87 MG tumor-bearing mice received an intravenous injection of about 1.85 MBq and 0.25 μg of ^111^In-DOTA-EB-cRGDfK. They were placed in metabolic cages individually for 8 days. Food and water were adequately provided. Urine and feces were continuously collected in each metabolic cage, weighed daily, and counted for radioactivity on a gamma counter. The %ID excreted at each time was thus determined. Cumulative excretion curves (urine and feces) were fitted with nonlinear regression by SigmaPlot 12.5 (Systat Software, Inc., San Jose, CA, USA).

### 4.7. Dosimetry

Dosimetry analysis of ^111^In-DOTA-EB-cRGDfK was based on biodistribution and excretion results and was performed as previously described [44,45]. The uptake and doses in various tissues/organs were derived from the radioactivity concentration in tissues and organs of interest, assuming a homogeneous distribution within each source region. The calculated mean value of % ID/g for the organs in mice was extrapolated to uptake in the organs of a 70 Kg adult. The extrapolated values (% IA) in the human organs at 2, 24, 48,72 and 96 h were fitted with exponential functions and integrated to obtain the number of disintegrations in the source organs; This information was entered into the OLINDA/EXM computer program. The International Commission on Radiological Protection (ICRP) 30 Gastrointestinal (GI) model and voiding urinary bladder model in OLINDA/EXM were used to estimate the number of disintegrations occurring in the excretory organs. this information was input into the OLINDA/EXM computer program. The integrals (MBqh/MBq administered) for organs were calculated and used for the dosimetry estimation.

### 4.8. Statistical Analysis

The results are expressed as mean and standard deviation (mean ± SD). The unpaired *t*-test was used for group comparisons. Data fitting and statistical analyses were computed using the SigmaPlot 12.5 (Systat Software, Inc., San Jose, CA, USA). Values of *p* < 0.05 were considered significant.

## 5. Conclusions

This study proposes a novel integrin-targeted molecule, DOTA-EB-cRGDfK, with long-term tumor retention ability. The results demonstrated that ^111^In-DOTA-EB-cRGDfK had higher accumulation in xenograft integrin-expressing U-87 MG tumor and that low imaging dose appears to be effective in treating tumors with high ^111^In-DOTA-EB-cRGDfK uptakes. It shows the possibility of being a therapeutic drug and is suitable for labeling diagnostic and therapeutic radionuclides. For future theranostic clinical use, ^111^In-DOTA-EB-cRGDfK/^177^Lu-DOTA-EB-cRGDfK has the potential to improve treatment efficacy using significantly lower quantities of ^177^Lu, and is a promising candidate for clinical translation to treat human brain cancer at the administered low dose. Further investigations with hematotoxicity, kidneys and intestine toxicities of administration are warranted.

## Figures and Tables

**Figure 1 ijms-22-05459-f001:**
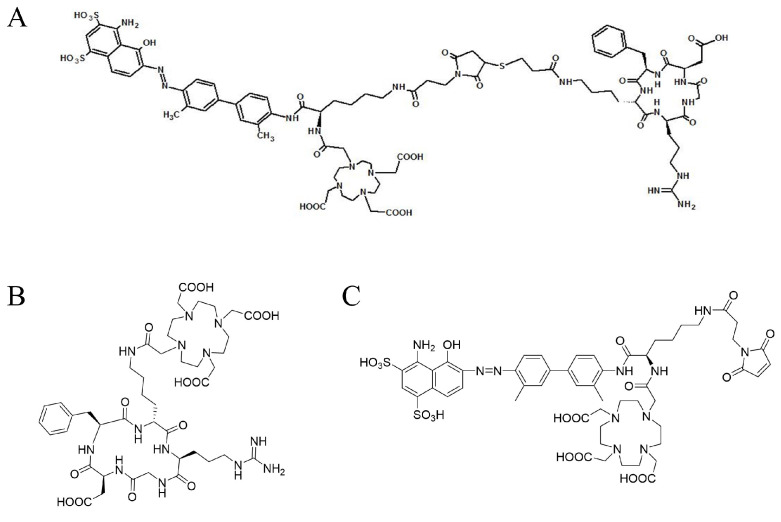
The structural formula of DOTA-EB-cRGDfK, DOTA-cRGDfK, and DOTA-EB. Molecular weights are 1900, 990.11, 1208.28, respectively.

**Figure 2 ijms-22-05459-f002:**
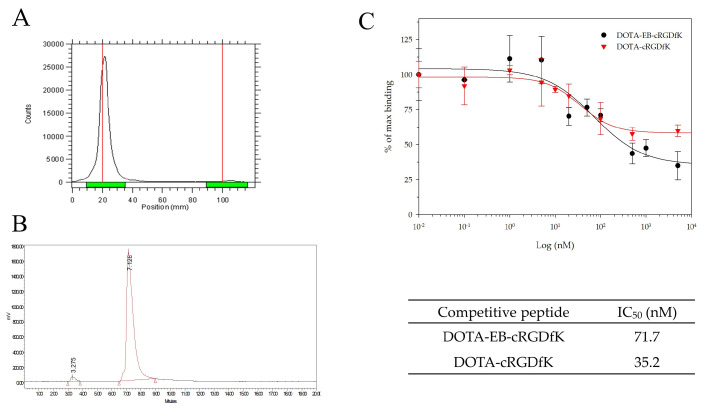
In vitro characterization of ^111^In-DOTA-EB-cRGDfK. (**A**) The labeling efficiency was 98% by radio-TLC analysis. (**B**) The radio HPLC analysis revealed that the radiochemical purity was 96% with a retention time of 7.2 min. (**C**) U-87 MG cells were used for cell-binding assays and competed with DOTA-EB-cRGDfK or DOTA-cRGDfK. The IC_50_ values of DOTA-EB-cRGDfK and DOTA-cRGDfK were 71.7 nM and 35.2 nM, respectively.

**Figure 3 ijms-22-05459-f003:**
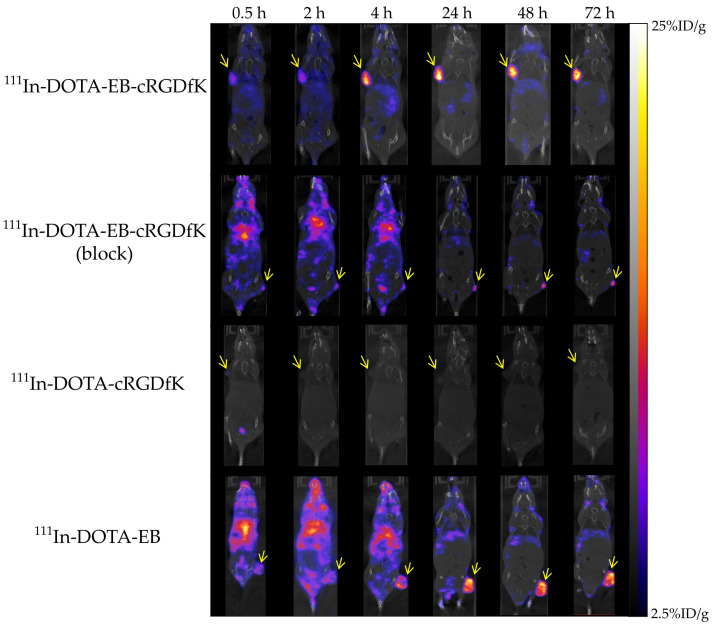
NanoSPECT/CT images of U-87 MG tumor-bearing mice. NanoSPECT/CT images at 0.5, 1, 4, 24, 48 and 72 h after injection of ^111^In-DOTA-EB-cRGDfK, ^111^In-DOTA-EB-cRGDfK with 120 times DOTA-EB-cRGDfK, ^111^In-DOTA-cRGDfK and ^111^In-DOTA-EB. Arrows indicate tumor location.

**Figure 4 ijms-22-05459-f004:**
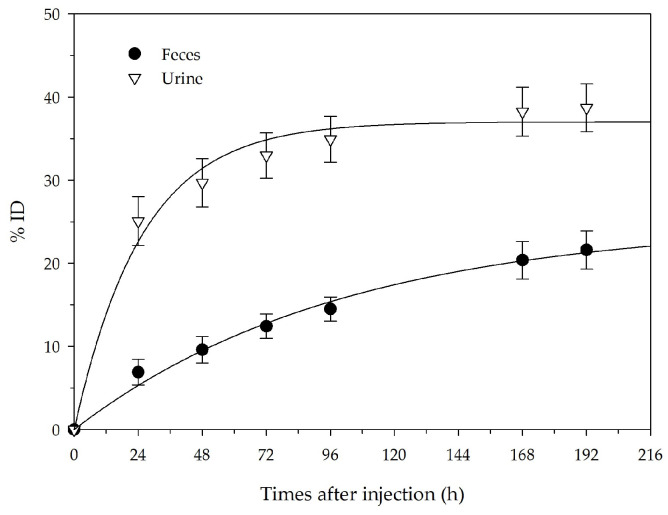
Cumulative urinary and fecal excretion of ^111^In-DOTA-EB-cRGDfK after the intravenous administration. Each point represents the mean ± SD for five animals.

**Table 1 ijms-22-05459-t001:** In vitro stability of ^111^In-DOTA-EB-cRGDfK at different times after incubation in normal saline or rat plasma at room temperature (mean ± SD, *n* = 3).

Incubation Time (h)	Rat Plasma (%)	Normal Saline (%)
0	96.8 ± 1.8	96.2 ± 2.2
2	95.7 ± 2.1	96.3 ± 1.7
4	96.3 ± 1.9	95.9 ± 1.8
24	95.8 ± 1.3	95.6 ± 1.8
48	96.2 ± 1.3	95.9 ± 0.8
96	96.3 ± 2.0	96.1 ± 1.0

**Table 2 ijms-22-05459-t002:** Biodistribution of ^111^In-DOTA-EB-cRGDfK after tail-vein injection in the U-87 MG tumor bearing mice.

Organ	2 h	24 h	48 h	72 h	96 h
Blood	13.46 ± 0.92	1.90 ± 0.16	1.44 ± 0.19	1.12 ± 0.38	0.88 ± 0.05
Skin	6.03 ± 0.7	5.80 ± 0.79	5.52 ± 0.84	5.56 ± 0.70	5.01 ± 1.03
Muscle	2.18 ± 0.27	1.19 ± 0.11	1.19 ± 0.14	1.26 ± 0.23	1.02 ± 0.14
Bone	1.95 ± 0.37	1.69 ± 0.56	1.38 ± 0.48	1.33 ± 0.70	0.96 ± 0.16
Brain	0.46 ± 0.09	0.32 ± 0.03	0.32 ± 0.08	0.30 ± 0.12	0.27 ± 0.03
Bladder	9.1 ± 1.55	7.86 ± 1.16	7.05 ± 1.50	6.53 ± 0.86	6.43 ± 2.03
Pancreas	2.42 ± 0.14	1.46 ± 0.13	1.38 ± 0.27	1.48 ± 0.20	1.25 ± 0.18
Spleen	9.73 ± 2.1	12.10 ± 1.84	12.88 ± 2.78	11.06 ± 2.85	8.83 ± 3.11
Stomach	5.72 ± 0.93	4.57 ± 0.63	4.14 ± 0.66	3.53 ± 0.70	2.17 ± 0.12
Small intestine	10.03 ± 1.05	7.01 ± 2.88	6.13 ± 2.4	6.91 ± 1.31	4.20 ± 2.13
Large intestine	4.91 ± 0.39	3.38 ± 0.58	2.75 ± 0.54	3.14 ± 1.01	1.67 ± 0.25
Bile	5.65 ± 6.39	2.32 ± 1.25	0.67 ± 0.51	3.49 ± 3.05	0.23 ± 0.13
Liver	7.79 ± 1.26	5.79 ± 0.37	6.15 ± 0.95	7.15 ± 2.51	5.70 ± 0.29
Kidney	13.34 ± 1.08	14.42 ± 0.38	13.41 ± 2.06	13.70 ± 2.73	11.01 ± 0.50
Heart	4.65 ± 0.27	2.79 ± 0.06	2.75 ± 0.55	2.62 ± 0.59	2.09 ± 0.21
Lung	7.11 ± 0.69	3.11 ± 0.28	3.92 ± 2.29	2.91 ± 0.31	2.92 ± 0.92
Tumor	19.44 ± 2.20	27.12 ± 2.70	26.53 ± 3.92	26.44 ± 6.16	18.55 ± 2.01
T/M ratio	8.92	22.79	22.29	20.98	18.19
T/B ratio	1.44	14.27	18.42	23.61	21.08

Values were presented as %ID/g, mean ± SD, (*n* = 4–5 at each time point). T/M, tumor/muscle; T/B, tumor/blood.

**Table 3 ijms-22-05459-t003:** Pharmacokinetic parameter estimates of ^111^In-DOTA-EB-cRGDfK and ^111^In-DOTA-cRGDfK after intravenous administration in U-87 MG tumor-bearing mice.

Parameter	Unit	^111^In-DOTA-EB-cRGDfK	^111^In-DOTA-cRGDfK
T_1/2__λ__z_	h	77.3	17.2
C_max_	% ID/mL	13.1	3.74
Cl	mL/h	0.426	12.9
AUC(_0→__∞_)	% ID/mL × h	242	4.02
MRT(_0→__∞_)	h	77.2	10.3

Calculated with the WinNonlin program for a noncompartmental model. T_1/2__λ__z_, terminal half-life; C_max_, maximum concentration; Cl, total body clearance; AUC, area under curve; MRT, mean residence time.

**Table 4 ijms-22-05459-t004:** Calculated organ radiation-dose estimates of ^111^In-DOTA-EB-cRGDfK for humans.

Organ	Estimated Dose (mSv/MBq)
Brain	0.026
Breasts	0.051
Gallbladder Wall	0.150
LLI Wall	0.622
Small Intestine	0.207
Stomach Wall	0.110
ULI Wall	0.376
Heart Wall	0.102
Kidneys	0.214
Liver	0.187
Lungs	0.096
Muscle	0.076
Ovaries	0.202
Pancreas	0.117
Red Marrow	0.080
Osteogenic Cells	0.176
Skin	0.043
Spleen	0.186
Thymus	0.070
Thyroid	0.057
Urinary Bladder Wall	0.195
Uterus	0.139
Total Body	0.086
Effective Dose	0.201

Dosimetry was analyzed by the OLINDA/EXM software. Extrapolated radiation dose for a 70-kg female adult. LLI, Lower large intestine; ULI, upper large intestine.

## Data Availability

Not applicable.

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
