# Peer review of "Molecular Imaging and Preclinical Studies of Radiolabeled Long-Term RGD Peptides in U-87 MG Tumor-Bearing Mice"

_ijms, 2021, doi:10.3390/ijms22115459_

Round 1

Reviewer 1 Report

The paper with the title “Molecular imaging and preclinical studies of radiolabeled long-term RGD peptides in U-87 MG tumor-bearing mice” describes the evaluation of an 111In-labeled RGD-peptide targeting integrins. The aim was to develop a stable radiotracer for cancer imaging and subsequent radionuclide therapy.

Comments:

Line 13: Authors should mention that the peptide was labelled with In-111.

Line 21: Which “other peptides” are meant here? Do they mean the control peptides which they also used? Authors should clarify and describe which peptides.

Line 49: What do the authors mean with “increase activity”? Authors should clarify this point.

Line 53: What is “100 times higher”? The inhibition potential? Authors should clarify this point.

Line 67: Authors should already here (not in the discussion) explain why EB was used for this study (e.g. effective moiety to improve EB-conjugated drug pharmacokinetic properties, prolongs blood circulation)

Line 72: Authors should mention here that they also developed two control peptides for comparison with the targeting peptide. They should also explain the concept of these two control peptides and why they were developed this way. This is important to understand the aim of the study.

Line 74: Authors should mention that the peptide was labelled with In-111 and why they are using In-111 as radiolabel.

Line 77: Authors should explain for what kind of tumors the peptide was developed. Since they are using are human glioblastoma cell line should it be used for glioblastomas? For the reader this is not clear and the purpose of this paper is missing.

Line 124: Authors should clarify „other peptides“ – they probably mean their “control peptides”?

Line 132: “111In-DOTA-EB was distributed throughout the body” – what do the authors mean? Please clarify.

Line 199-208: This part of the discussion explains the main aim of the study. Therefore this section should be moved to the introduction (and also parts of it to the abstract) so that the reader understands the purpose of the study already when starting to read the paper.

Line 225: What do the authors mean with “many non-specific bindings”? This should be clarified.

Minor changes:

Line 12: Change „tumor new blood vessels” to “new tumor blood vessels”

Line 38: Change „tumor new blood vessels” to “new tumor blood vessels”

Line 49: Change “as increase activity” to “as to increase activity”

Line 50: Change “and so on” to “etc.”

Line 63: Change “is used” to “was used”

Line 64: Change “Many RGD-derived drugs have entered clinical trials and are used to diagnose cancer, such as 68Ga-NOTA-BBN-RGD (Phase 1), 18F-RGD-K5 (Phase 2), 18F-Al-NOTA-PRGD2 ((Phase 1) and 18F-FPPRGD2 (Phase 2), etc.” to “Many RGD-derived drugs have entered clinical trials, such as 68Ga-NOTA-BBN-RGD (Phase 1), 18F-RGD-K5 (Phase 2), 18F-Al-NOTA-PRGD2 ((Phase 1) and 18F-FPPRGD2 (Phase 2), and are used to diagnose different types of cancer.”

Line 69: Change “peptide that binds integrin” to “peptide that binds to integrins”

Line 70: Change “labeling β-emitting” to “labelling with the β-emitting”

Line 82: Change “after radiolabeled” to “after radiolabeling”

Line 83: Change “95% by” to “95% determined by”

Line 111: Change “radio-purity” to “radiochemical purity”

Line 126: Change “When blocking with 120 times DOTA-EB-cRGDfK” to “when blocking with a 120fold excess of DOTA-EB-cRGDfK”

Line 138: Change “with 120 times DOTA-EB-cRGDfK” to “with a 120fold molar excess of DOTA-EB-cRGDfK”

Line 278: Change “100 ug” to “100 µg”

Line 296: Change “The HPLC was measured by UV detector (220 nm)” to “HPLC-analysis was performed using UV detection at 220 nm and a radio detector with a ZORBAX SB-C18 column”

Line 302: Change “A concentration ranges from 0.01 nM to 5000 nM of DOTA-EB- 302

cRGDfK or DOTA-cRGDfK was added to the wells” to “Different concentrations ranging from 0.01 nM to 5000 nM of DOTA-EB-cRGDfK or DOTA-cRGDfK were added to the wells”

Line 323: Change “with 120 times (300 μg)” to “with a 120fold molar excess (300 µg)”

Line 383: Authors should rewrite this sentence.

Author Response

Reviewer 1: Comments and Suggestions for Authors

The paper with the title “Molecular imaging and preclinical studies of radiolabeled long-term RGD peptides in U-87 MG tumor-bearing mice” describes the evaluation of an 111In-labeled RGD-peptide targeting integrins. The aim was to develop a stable radiotracer for cancer imaging and subsequent radionuclide therapy.

Response: The authors would like to thank Reviewer 1 for all of the insightful and helpful comments and suggestions. Our responses can be found below.

Major comments:

  1. Line 13: Authors should mention that the peptide was labelled with In-111.

Response: This point has been clarified in the Abstract section.

  1. Line 21: Which “other peptides” are meant here? Do they mean the control peptides which they also used? Authors should clarify and describe which peptides.

Response: We mean the control peptides (111In-DOTA-cRGDfK and 111In-DOTA-EB) and this point has been clarified in the Abstract section.

  1. Line 49: What do the authors mean with “increase activity”? Authors should clarify this point.

Response: We mean the binding affinity and this point has been clarified in the Introduction section.

  1. Line 53: What is “100 times higher”? The inhibition potential? Authors should clarify this point.

Response: Response: We mean the binding affinity of cRGDfK to integrin αvβ3 is much higher than the linear RGD peptide. The sentence has been modified in the Introduction section.

  1. Line 67: Authors should already here (not in the discussion) explain why EB was used for this study (e.g. effective moiety to improve EB-conjugated drug pharmacokinetic properties, prolongs blood circulation)

Response: The micromolar affinity and reversible binding of EB derivatives to albumin extend the half-life of the drug in the blood. The EB-conjugated drug can improve pharmacokinetic properties and prolongs blood circulation. This part was added in the Introduction section.

  1. Line 72: Authors should mention here that they also developed two control peptides for comparison with the targeting peptide. They should also explain the concept of these two control peptides and why they were developed this way. This is important to understand the aim of the study.

Response: We developed two control peptides for comparison with the targeting peptide. DOTA-cRGDfK is the current standard of RGD peptides, which is used to compare the tumor-targeting effects. The purpose of DOTA-EB is to understand whether EB has a binding or modulating effect on tumors. The concept of these two control peptides was added in the Introduction section.

  1. Line 74: Authors should mention that the peptide was labelled with In-111 and why they are using In-111 as radiolabel.

Response: Indium-111 was used because it emits gamma rays (171 and 245 keV), with a long half-life (2.8 days), and is suitable for long-term imaging. This point has been clarified in the Introduction section.

  1. Line 77: Authors should explain for what kind of tumors the peptide was developed. Since they are using are human glioblastoma cell line should it be used for glioblastomas? For the reader this is not clear and the purpose of this paper is missing.

Response: We selected U-87MG human glioblastoma with a high expression of αvβ3 integrin as the model. And DOTA-EB-cRGDfK is used for glioblastoma. This point has been clarified in the Introduction and Abstract section.

  1. Line 124: Authors should clarify „other peptides“ – they probably mean their “control peptides”?

Response: We mean control peptides. This point has been clarified in the Result section.

  1. Line 132: “111In-DOTA-EB was distributed throughout the body” – what do the authors mean? Please clarify.

Response: What we mean is that 111In-DOTA-EB showed relatively strong radiation signals in non-tumor organs such as muscle and liver. This point has been clarified in the Result section.

  1. Line 199-208: This part of the discussion explains the main aim of the study. Therefore this section should be moved to the introduction (and also parts of it to the abstract) so that the reader understands the purpose of the study already when starting to read the paper.

Response: We have reflected this comment by modifying the Introduction and Discussion section.

  1. Line 225: What do the authors mean with “many non-specific bindings”? This should be clarified.

Response: This point is related to point 10. We have therefore amended the text.

Minor comments:

Line 12: Change „tumor new blood vessels” to “new tumor blood vessels”

Line 38: Change „tumor new blood vessels” to “new tumor blood vessels”

Line 49: Change “as increase activity” to “as to increase activity”

Line 50: Change “and so on” to “etc.”

Line 63: Change “is used” to “was used”

Line 64: Change “Many RGD-derived drugs have entered clinical trials and are used to diagnose cancer, such as 68Ga-NOTA-BBN-RGD (Phase 1), 18F-RGD-K5 (Phase 2), 18F-Al-NOTA-PRGD2 ((Phase 1) and 18F-FPPRGD2 (Phase 2), etc.” to “Many RGD-derived drugs have entered clinical trials, such as 68Ga-NOTA-BBN-RGD (Phase 1), 18F-RGD-K5 (Phase 2), 18F-Al-NOTA-PRGD2 ((Phase 1) and 18F-FPPRGD2 (Phase 2), and are used to diagnose different types of cancer.”

Line 69: Change “peptide that binds integrin” to “peptide that binds to integrins”

Line 70: Change “labeling β-emitting” to “labelling with the β-emitting”

Line 82: Change “after radiolabeled” to “after radiolabeling”

Line 83: Change “95% by” to “95% determined by”

Line 111: Change “radio-purity” to “radiochemical purity”

Line 126: Change “When blocking with 120 times DOTA-EB-cRGDfK” to “when blocking with a 120fold excess of DOTA-EB-cRGDfK”

Line 138: Change “with 120 times DOTA-EB-cRGDfK” to “with a 120fold molar excess of DOTA-EB-cRGDfK”

Line 278: Change “100 ug” to “100 µg”

Line 296: Change “The HPLC was measured by UV detector (220 nm)” to “HPLC-analysis was performed using UV detection at 220 nm and a radio detector with a ZORBAX SB-C18 column”

Line 302: Change “A concentration ranges from 0.01 nM to 5000 nM of DOTA-EB- 302

cRGDfK or DOTA-cRGDfK was added to the wells” to “Different concentrations ranging from 0.01 nM to 5000 nM of DOTA-EB-cRGDfK or DOTA-cRGDfK were added to the wells”

Line 323: Change “with 120 times (300 μg)” to “with a 120fold molar excess (300 µg)”

Line 383: Authors should rewrite this sentence.

Response: Thanks for reviewer’s comments, we have corrected all minor comments above.

Reviewer 2 Report

Dear Authors,

Frist of all, I would like to remind you that the use of animals in experimental studies should be permitted (provide permission number and the name of the ethics organization that gave you the permission), justified and supported by the pre-eliminary results with very promissing results. Such data are not provided in the paper I received.

After careful analysis of the presented data, I do not find great adavateges in the use of  111In-DOTA-EB-cRGDfK in molecular immagining, neither targeted radionuclide therapy; respect to the currently used (and under investigation) technology.  In addition, the comparison of the presented here results with other known technlogies is missing.

Best regards

Reviewer

Author Response

Reviewer 2: Comments and Suggestions for Authors

Frist of all, I would like to remind you that the use of animals in experimental studies should be permitted (provide permission number and the name of the ethics organization that gave you the permission), justified and supported by the preliminary results with very promising results. Such data are not provided in the paper I received.

Response: The authors would like to thank Reviewer 2 for all of the insightful and helpful comments and suggestions. The experimental protocol was approved by the Institutional Animal Care and Use Committee of Institute of Nuclear Energy Research (approval ID: 107169). The approval ID was added in the Materials and Methods and Institutional Review Board Statement.

After careful analysis of the presented data, I do not find great advantages in the use of 111In-DOTA-EB-cRGDfK in molecular imaging, neither targeted radionuclide therapy; respect to the currently used (and under investigation) technology.  In addition, the comparison of the presented here results with other known technologies is missing.

Response: We have added a paragraph about the advantages of DOTA-EB-cRGDfK in the Discussion section. Briefly, it shows higher tumor uptake, extended half-life and theranostic potential. And according to our unpublished preliminary data, 177Lu-DOTA-EB-cRGDfK also show high uptake in human glioblastoma.

Reviewer 3 Report

In this manuscript M. Li  and colleagues explored the use of a novel peptide construct for theranostic application in preclinical study (Xenograft mice model).

The study is well conducted, well presented, easy to follow and very promising.

Here are some points that the authors may consider:

Major points:

This study is conducted on glioblastoma cell I wonder why the authors have not chosen another type of cancer cell line that are more relevant to the peptide used. I guess the peptide do not cross the blood-brain barrier therefore this peptide should have been tested with another type of cells. This should be mentioned in the discussion and explained. If this is only a proof of concept, it should be mentioned that other experiments are ongoing with a more relevant type of cells regarding the peptide.

Minor points:

-Line 21: “than other peptide”. Which other peptides ?

-Line 52-54: this is not clear and should be reformulated: 100 times higher than what? The adhesion of fibronectin to cells regarding which peptide ?

-Line 67: please explain the mechanism of albumin – binding of EB and why it brings stability to the peptide (it is more or less explain in the discussion, but should better explained in this section

-Line 84; explain Rf

-Line 208: than instead of then

-Line 219: explain the mechanism of EB-albumin (as asked above)

-Line 231: what does “enhanced permeability” mean in this context ? does the peptide has an effect on the plasma membrane ? does the peptide enter the cells ? or it stays at the surface bound to the receptors ?

-Line 233-235: Please explain this sentence what does “they” refers to ?

-Line 257: rewrite the sentence: “a first-in-human study of has.”

-Line 383: rewrite this sentence please.

-Line 391: ..is a promising candidate

Author Response

Reviewer 3: Comments and Suggestions for Authors

In this manuscript M. Li and colleagues explored the use of a novel peptide construct for theranostic application in preclinical study (Xenograft mice model).

The study is well conducted, well presented, easy to follow and very promising.

Response: The authors would like to thank Reviewer 3 for all of the insightful and helpful comments and suggestions. Our responses can be found below.

Major comments:

This study is conducted on glioblastoma cell I wonder why the authors have not chosen another type of cancer cell line that are more relevant to the peptide used. I guess the peptide do not cross the blood-brain barrier therefore this peptide should have been tested with another type of cells. This should be mentioned in the discussion and explained. If this is only a proof of concept, it should be mentioned that other experiments are ongoing with a more relevant type of cells regarding the peptide.

Response:  This point has been clarified in the Introduction and Discussion section. U-87 MG has a high expression of αvβ3 integrin and has higher tumor uptake compare with our unpublished preliminary data in breast cancer MDA-MB-231 (25.09 ± 4.76%ID/g and 7.19 ± 0.45%ID/g at 24 h, respectively). Regarding your concerns about the blood-brain barrier, it may be defective in malignant gliomas. Radiolabeled peptides may work because BBB breakdown is common in high-grade gliomas and brain metastases. And we have added this point in the Discussion section.

Ref:

  1. Schneider, Stefan & Ludwig, Thomas & Tatenhorst, Lars & Braune, Stephan & Oberleithner, Hans & Senner, Volker & Paulus, Werner. (2004). Glioblastoma cells release factors that disrupt blood-brain barrier features. Acta neuropathologica. 107. 272-6.
  2. Belykh, E.; Shaffer, K. V.; Lin, C.; Byvaltsev, V. A.; Preul, M. C.; Chen, L., Blood-Brain Barrier, Blood-Brain Tumor Barrier, and Fluorescence-Guided Neurosurgical Oncology: Delivering Optical Labels to Brain Tumors. Frontiers in oncology 2020, 10, 739.

Minor comments:

-Line 21: “than other peptide”. Which other peptides ?

Response: We mean the control peptides (111In-DOTA-cRGDfK and 111In-DOTA-EB) and this point has been clarified in the Abstract section.

-Line 52-54: this is not clear and should be reformulated: 100 times higher than what? The adhesion of fibronectin to cells regarding which peptide ?

Response: We mean the binding affinity of cRGDfK to integrin αvβ3 is much higher than the linear RGD peptide. The sentence has been modified in the Introduction section.

-Line 67: please explain the mechanism of albumin – binding of EB and why it brings stability to the peptide (it is more or less explain in the discussion, but should better explained in this section

Response: The micromolar affinity and reversible binding of EB derivatives to albumin extend the half-life of the drug in the blood. The EB-conjugated drug can improve pharmacokinetic properties and prolongs blood circulation. The part was added in the Introduction section.

-Line 84; explain Rf

Response: Relative front (Rf) is defined as the ratio of distance traveled by the component to the distance traveled by the solvent front from the sample spot. This point has been added in the Materials and Methods section.

-Line 208: than instead of then

Response: Thanks for reviewer’s comments, we have corrected.

-Line 219: explain the mechanism of EB-albumin (as asked above)

Response: This point is related to above point. We have therefore amended the text.

-Line 231: what does “enhanced permeability” mean in this context ? does the peptide has an effect on the plasma membrane ? does the peptide enter the cells ? or it stays at the surface bound to the receptors ?

Response: Thanks for reviewer’s comments and suggestions. The enhanced permeability and retention (EPR) effect is the property of the drug accumulate more in tumor than in normal tissues, and may be due to hypervascularization. We use this concept to explain the tumor accumulation of 111In-DOTA-EB. EB may bind strongly to albumin, extravasates, and remains for a prolonged time in the extravascular space due to the EPR effect of tumors. we will study more peptide binding details in the future.

Ref: Yao, L.; Xue, X.; Yu, P.; Ni, Y.; Chen, F., Evans Blue Dye: A Revisit of Its Applications in Biomedicine. Contrast media & molecular imaging 2018, 2018, 7628037.

-Line 233-235: Please explain this sentence what does “they” refers to ?

Response: We have rewritten this sentence.

-Line 257: rewrite the sentence: “a first-in-human study of has.”

Response: We have rewritten this sentence.

-Line 383: rewrite this sentence please.

Response: We have rewritten this sentence.

-Line 391: ..is a promising candidate

Response: Thanks for reviewer’s comments, we have corrected.

Round 2

Reviewer 2 Report

I have no more comments